# Wideband Array Signal Processing with Real-Time Adaptive Interference Mitigation

**DOI:** 10.3390/s23146584

**Published:** 2023-07-21

**Authors:** Adam Whipple, Mark W. Ruzindana, Mitchell C. Burnett, Jakob W. Kunzler, Kayla Lyman, Brian D. Jeffs, Karl F. Warnick

**Affiliations:** 1Electrical & Computer Engineering, Brigham Young University, Provo, UT 84602, USA; 2Breakthrough Listen Project, U.C. Berkeley, Berkeley, CA 74707, USA; 3Naval Information Warfare Center, Charleston, SC 29401, USA

**Keywords:** phased array antenna, wideband beamforming, digital signal processing, radio frequency interference mitigation

## Abstract

Wideband beamforming and interference cancellation for phased array antennas requires advances in signal processing algorithms, software, and specialized hardware platforms. A high-throughput array receiver has been developed that enables communication in radio frequency interference-rich environments with field programmable gate array (FPGA)-based frequency channelization and packetization. In this study, a real-time interference mitigation algorithm was implemented on graphics processing units (GPUs) contained in the data pipeline. The key contribution is a hardware and software pipeline for subchannelized wideband array signal processing with 150 MHz instantaneous bandwidth and interference cancellation with a heterogeneous, distributed, and scaleable digital signal processing (DSP) architecture that achieves 30 dB interferer cancellation null depth in real time with a moving interference source.

## 1. Introduction

Radar, radio astronomy, and terrestrial and satellite communication systems are increasingly moving from single-antenna transceivers to phased array antenna front ends. Active array sensor provide high gain, sidelobe control, and flexibility in the beam pattern. Digital beamformer back ends can form multiple simultaneous beams with electronically scanned pointing directions. A significant benefit of array receivers with a digital beamformer is the ability to provide adaptive radio frequency interference (RFI) mitigation to reduce in-band hostile or unintentional interferers.

The challenge for phased array systems with digital signal processing is primarily one of throughput. The system’s computational burden scales with the bandwidth, number of array elements, and the complexity of the signal processing algorithms. To overcome the throughput requirements for wideband array signal processing, scaleable, heterogeneous system architectures using field programmable gate arrays (FPGAs), high-performance computing (HPC) hardware, and graphics processing units (GPUs) have been developed [1,2,3,4]. Advancements in FPGA technologies and increased availability of commercial HPC hardware and powerful network switches have enabled a multi-stage architecture with each hardware element optimized for a given signal processing pipeline task [5].

Compared to designs that use only FPGAs or dedicated DSP hardware, a heterogeneous array signal processing architecture reduces the overall software development burden and distributes the computational workload across multiple nodes [6]. FPGAs are well-suited to regular structures that are needed for the first computational block after digitization in a large array receiver, an FFT, or polyphase filterbank for frequency channelization. Complex matrix-based signal processing algorithms are more difficult to implement using FPGAs. Even with advanced development tools, the cost of FPGA block development and debugging can exceed that of the hardware. Performing complex signal processing function in GPUs rather than FPGAs can significantly reduce development time. Moving complex functions to GPUs allows the use of general purpose programming languages that are much better suited to matrix-based processing. The challenge of processing large bandwidths can be overcome by using a network switch to distribute data to HPCs hosting GPUs for beamforming or correlation operations and evenly distributing the processing load across multiple distributed nodes [7,8].

The heterogeneous FPGA/HPC/GPU architecture has been used to develop scaleable, distributed signal processing architectures for phased array beamforming in applications to astronomical imaging [1,2]. Heterogeneous systems are used by astronomical receivers such as the Large-aperture Experiment to Detect the Dark Age (LEDA) [9], Focal L-band Array for the Green Bank Telescope (FLAG) [10] and the Advanced L-band Phased Array Camera for Astronomy (ALPACA) [11].

Our goal is extend the scaleable, heterogeneous array signal processing architecture by adding the capability for real-time interference mitigation. Interference mitigation is a topic of significant interest in the array signal processing community for RFI reduction in wireless communications systems and removing human-caused RFI in imaging systems for radio astronomy [12,13,14,15,16]. Beamforming can be used to increase a channel’s secrecy or satisfy the network demands of internet-of-things devices [17,18,19]. Adaptive phased array beamforming has long been used for RFI mitigation, but challenges remain for applications that require simultaneous real-time rejection of interfering signals in different sub-band channels over a wide bandwidth. Adaptive array RFI cancellation methods place pattern nulls on interfering signals or otherwise remove unwanted signals for anti-jamming and mitigation of RFI from many sources [12,13,14,15,16]. As the number of array elements, formed beams, and instantaneous processing bandwidth increases, so does the computational burden and IO demand on the digital back end. As a result of this, RFI cancellation can be limited to a narrow bandwidth or a limited number of formed beams.

This paper provides experimental results that demonstrate the real-time RFI cancellation of narrowband interferers in filterbank output subchannels within a system that processes a wide overall bandwidth. This effort builds on previous work [2,9,10,11] by adding the capability to perform subspace projection to remove narrowband RFI and adapting the architecture to communications applications. Preliminary versions of the system used in this work have been reported in [20,21,22].

The main contribution of this work is to demonstrate that a heterogeneous FPGA, HPC, and GPU-based system can process signals from an array with a wide bandwidth of 150 MHz with frequency channelization and beamforming, while simultaneously performing real-time cancellation in subbands that can be arbitrarily selected from the overall system instantaneous bandwidth. We also provide an analysis of the tradeoff between the length of the array signal correlation time and the accuracy with which a moving interfering signal can be identified and the null depth with which the RFI can be removed from the array output signal. This is used to determine the optimal array correlation time used in the RFI cancellation algorithm implementation, which allows an interference cancellation null depth of 30 dB to be achieved for a moving RFI source.

The system consists of an antenna array, multichannel analog receiver, digitizers, FPGAs, networked data transmission, and a cluster of HPCs with GPUs. The analog–to–digital interface is implemented using an FPGA platform with many multi-gigabit transceivers for interfacing with high-speed analog–to–digital converters (ADCs) and networked data transmissions. The programmable fabric of the FPGA is used for frequency channelization and data packetization. Sampled and channelized signals from the sensor array are transmitted over Ethernet to a second stage. The second stage processor is a cluster of HPCs with GPUs for array signal correlation and real-time interference cancellation.

In an application system, the wideband signal including all channelizer outputs would be passed on to a communications receiver that extracts data from the signal of interest. The interference cancellation demonstrated in this work is done individually within subchannels produced by the FFT channelizer. As interferers are generally narrowband in our applications of interest, the cancellation technique demonstrated in the paper would be replicated across the channelizer outputs and the interferers within each subchannel are mitigated individually.

In Section 2, we describe the system architecture in detail. Section 3 provides an overview of the RFI mitigation algorithm. Section 4 gives experimental results for real-time interference cancellation in a wideband phased array receiver.

## 2. System Overview

The heterogeneous approach to phased array beamforming and signal processing has been implemented for wideband phased array communications with real-time adaptive interference cancellation. The system consists of a 16-element phased array operating at the X band, analog signal handling, and a digital back end consisting of FPGAs, Ethernet switch, HPCs, GPUs, and solid state drives (SSDs) for data storage. A block diagram of the multi-element data acquisition system architecture can be seen in Figure 1. The digital signal processing system performs real-time beamforming, array output voltage correlation, RFI mitigation using subspace projection (SP), and a diagnostic mode that allows the capture of raw sampled array output voltages. A software interface distributes the processes needed for these functions across multiple HPCs.

The analog front end supports an instantaneous bandwidth of 150 MHz for each of the 16 microstrip patch antenna elements. Signal handling consists of low noise amplifiers (LNAs), downconverters from a 10.2 GHz RF band center frequency to a 300 MHZ IF. Antialiasing filters select 150 MHz for baseband subsampling in the second Nyquist zone at a rate of 400 Msamples/s. The analog receiver chain has 66 dB of gain and 1150 K noise temperature (7 dB noise figure).

The first stage of the digital back end consists of three Smart Network ADC Processor (SNAP) boards, and the second stage utilizes four Tyan HPCs. The SNAP board was designed by CASPER (Collaboration for Astronomy Signal Processing and Electronics Research) [23] and is equipped with three Analog Devices, HMCAD1511 ADCs, a Xilinx Kintex-7 FPGA (XC7K160T-2FFG676C), and two 10 GbE interfaces. The Tyan HPC has a dual CPU and PCIe bus architecture with each HPC containing two GeForce RTX 2080 Ti GPUs and two 10 GbE network interface cards (NICs). An Arista DCS-7050QX-32-R Ethernet switch is used to facilitate the transport of the sampled data packets between SNAP boards and GPU HPCs.

The firmware design running on the SNAP board and performing digitization and frequency channelization was designed using the CASPER development environment [24]. Frequency channelization allows beamforming and array signal processing for the wideband receiver to be performed in subbands narrow enough to avoid beam squint. The digitization rate is 400 Msamples/s. The sampled array signals are channelized in frequency using a 256 point fast Fourier transform (FFT). Of the frequency channelized outputs, 32 subbands are discarded to remove antialiasing filter transition bands and 96 subbands each 1.5625 MHz wide are retained, corresponding to an operating bandwidth of 150 MHz. Channelized samples are grouped into 12 packets of 8 frequency bins and 85 time windows for the next signal processing stage.

Packets containing data from each of six array ports per SNAP board are transferred from the SNAP boards to a port on each HPC for beamforming. Packet transfer is accomplished by 10 GbE cores on the SNAP FPGAs to the network switch which transfers the data to HPCs, dividing the bandwidth evenly over the GPUs. Packet sockets are used to bypass the kernel for low-latency, high-throughput communication. The packets from the network switch are captured and processed in the HPC with network communications thread software that utilizes the high-availability shared pipeline engine (HASHPIPE) framework [4,25]. HASHPIPE uses rotating circular buffers to transfer data between multiple threads. A GPU accelerator thread is used to perform array signal processing computations in real time. To minimize GPU calls, the data are grouped into 50 sets of 85 time samples. This means one packet of 4250 time samples is used to estimate the weights to remove the RFI. A data storage save thread is used to write the data to binary files that are saved to a redundant array of independent disks (RAID) consisting of SSDs. To maximize write speed, RAID 0 mode is used [26] in the data storage system. This enables the system to write at up to 4.16 Gbps per instance on the HPC network.

In some applications, the time delay from input to output or latency is important. For our system to run in real time, it has to be able to process the data within 2.7 ms; that is, the time corresponding to the 4250 time samples processed together. The timing for the GPU threads used for the interference calculation is 1.4 ms for the operational mode that only performs beamforming and 2.3 ms for the mode that performs the RFI cancellation. Even with the added RFI cancellation step, the latency is similar to other reported heterogeneous array signal processing systems.

## 3. Subspace Projection for Interference Cancellation

To remove RFI from received signals, temporal or spatial filters can be used. Time domain interference cancellation methods such as the least mean square (LMS) technique adaptively remove noise and interference by minimizing an error signal. Other time–domain methods use a predictive model for an interfering signal to minimize RFI. Spatial methods generally place a null on RFI in the response pattern of a sensor array. The technique used in this work is based on interference subspace identification and removal of the unwanted signal by subspace projection [3]. Figure 2 shows the implementation of the algorithm in the system.

In the subspace projection algorithm, the sample-estimated array output voltage correlation matrix $\hat{R}$ of the received data during one short time integration (STI) period is decomposed to its eigenvector and eigenvalue components using
(1)$${\hat{R}}_{k}=U_{k}\Lambda_{k}U_{k}^{-1}$$

The interference is assumed to be the strongest received signal, so we can partition **U** as $\left[u_{d,k}|U_{s+z}\right]$, where $u_{d,k}$ corresponds to the largest eigenvalue of the signal correlation matrix. With this partition, we create the projection operator
(2)$$P_{\mathrm{SP}}=I-u_{d,k}u_{d,k}^{H}$$

This projection is applied to the nominal beamformer weighting coefficients before the beamformer array signal processing step using
(3)$$w_{\mathrm{SP}}=P_{\mathrm{SP}}w$$

The real-time beamformer with these modified weights forms a beam steered to a signal of interest and removes the interferer subspace.

Subspace projection is based on the dominant eigenvectors of the array output voltage correlation matrix and the modified beamformer places a null in the mean square direction, combining the strongest signals received by the array. If the RFI is significantly stronger than the signal of interest (SoI), the RFI subspace is removed from the array output by the beamformer. A limitation of subspace projection is that if the SoI is stronger than the RFI or above the noise floor (which is normal for communications but uncommon in passive sensing applications), the spatial null may be placed on the SoI instead of the interference. In cases where the RFI power level is close to that of the SoI instead of being significantly higher, more sophisticated subspace partitioning techniques are required.

A key challenge with adaptive RFI mitigation is that interferer motion reduces the achievable null depth due to subspace smearing [27,28]. For long STI lengths, interferer motion leads to a poor estimated interferer subspace in (Equation 2). Short STI lengths lead to less subspace smearing, but fewer samples in the correlation matrix averaging step leads to high sample estimation error.

Numerical simulations were used to understand the impact of subspace smearing on RFI cancellation and determine the optimal range of STI lengths. The modelled array corresponded to one four-element row of the 4 × 4 array used in experimental tests. The signal model included thermal noise, a moving interferer, and a tone representing a signal of interest. The SoI and interferer were both tones at 150 and 171 MHz, respectively. The interference ranged from 3 to 27 dB stronger than the SoI. As shown in Figure 3, the integration time must be small enough that the interferer motion is limited. The optimal STI length depends on the power of the interferer, with a weaker interferer requiring more time to obtain a better subspace estimate.

In the experimental results, after frequency channelization, correlated array voltages are averaged over 4250 time samples, corresponding to an STI time of 2.7 ms. The interferer motion rate used in the tests was roughly 0.01 degrees per short time integration, corresponding to a motion rate of 0.0003 HPBW per STI. The interferer motion rate per STI is near the optimal value in Figure 3 for an INR before cancellation of 35–40 dB.

## 4. Experimental Results

For experimental tests, the subspace projection algorithm was applied to the samples in real time using the HASHPIPE data pipeline framework and and NVidia CUDA packages for GPU programming. To process the large system bandwidth in real time, parallel programming was required. Since beamforming is performed after a coarse FFT channelizer, this is a narrowband beamformer that is applied separately in each coarse channel. The system output signal, interferer, and noise spectra are produced with a fine FFT on the stored beamformer system output in post-processing.

The adaptive beamformer was tested with a moving RFI source produced by a signal generator and movable 4 × 4 X band microstrip patch antenna array. The SoI was stationary and at the array boresight. The nominal beamformer before subspace projection had unity weighting coefficients. The RFI source was placed at 45 degrees from the array boresight and was moved during the scan. The scan duration was 6 s and the RFI moved approximately 20 degrees over this time. The RFI transmitter was returned to the same location and the same movement was performed for both the real-time beamformer (RTBF) mode and the subspace projection mode (XRFI) so that a comparison could be made.

The system output spectrum is shown in Figure 4. The results for XRFI mode indicate the cancellation of the RFI while retaining the SoI. For ease of observing the results, the RFI was placed at a coarse bin center and the SoI was placed within the same coarse bin. The signal at 10.2 GHz is the downconverter local oscillator (LO) frequency and is the center of the system bandwidth. The tones near 10.22 GHz and 10.255 GHz are not present in the analog array receiver outputs and are likely aliased signals introduced in the digital stage which could be removed using additional signal processing if needed. Spectral leakage around the SoI and RFI is due to the SoI not being placed at a bin center of the FFT coarse channelizer. Figure 5 shows an expanded view with a narrower range of spectra around the RFI and SoI tones (SoI at 10.19355 GHz and the RFI near 10.19375 GHz). The SNR for this experiment was about 25 dB and the INR was set to 35 dB. The interferer null depth can be seen to be 30 dB.

While narrowband tones are convenient to demonstrate RFI cancellation, in practice, interfering signals are modulated and have wider bandwidth. To demonstrate the capability of subspace projection to cancel modulated RFI, the system was tested with an FM-modulated RFI source. The SoI and the center frequency for the RFI were kept the same as the previous experiment with the moving interferer. The RFI modulation bandwidth was 50 kHz. Results are shown in Figure 6. The RFI and SoI are contained within the same coarse frequency bin. It can be seen in Figure 7 that the modulated RFI is removed in the processed system output.

## 5. Conclusions

We have presented a real-time adaptive beamforming sensor array based on a heterogeneous architecture, capable of processing a wideband signal while rejecting interference using the subspace projection spatial filtering algorithm. Interference cancellation is implemented using a GPU thread that processes one FFT channelizer output. By replicating the cancellation process, separate interferers can be suppressed in multiple subchannels, leading to a wideband RFI-free signal that can be passed on to a communications receiver. The subchannel-based approach allows the RFI cancellation to be combined with the FFT channelizer, used for orthogonal frequency division multiplexing (OFDM) and other subchannel-based schemes commonly used in communication systems. The system architecture is currently being used for radio astronomy and wireless communications and could be used in other fields that require real-time beamforming for large antenna arrays. Real-time signal processing including an RFI mitigation algorithm implemented on GPUs will enable wireless communication in interference-rich environments over a wide operating bandwidth.

## Figures and Tables

**Figure 1 sensors-23-06584-f001:**
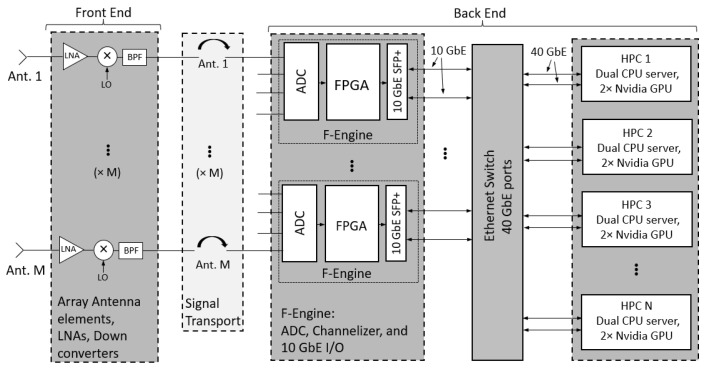
Wideband communications phased array receiver system diagram. Communications signals, noise, and interference are received by an antenna array. Analog signals are amplified, downconverted, filtered for antialiasing, and sampled using analog to digital converters. Sampled data are processed using an FPGA for frequency channelization on FPGAs. Channelized samples are packetized and transferred to an array of HPCs and GPUs for beamforming and real-time interference cancellation, leaving an interference-free wideband signal for use in a wideband communications application.

**Figure 2 sensors-23-06584-f002:**
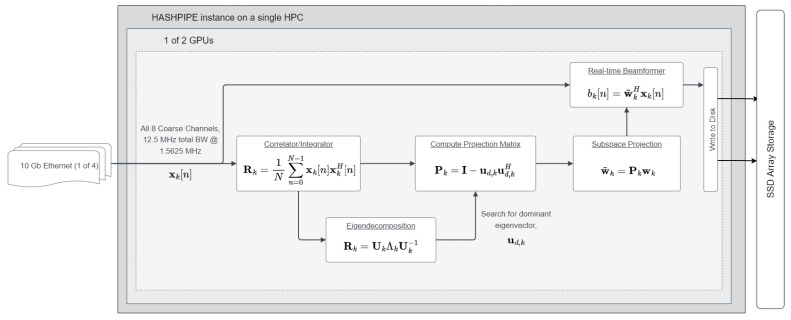
Functional diagram of the system including the subspace projection algorithm. The received signal is correlated and then decomposed into its eigenvector and eigenvalue components. The strongest eigenvector is assumed to be a representation of the RFI. A projection is created using this eigenvector and applied to the signal to remove the interferer.

**Figure 3 sensors-23-06584-f003:**
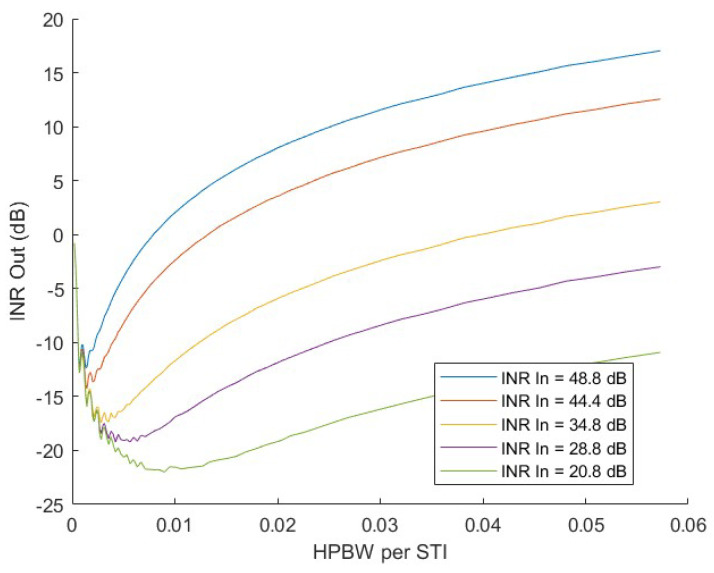
Modeled subspace projection results for a four-element linear array with moving interferer and subspace projection. The interference to noise ratio (INR) after subspace projection is computed for varying short time integration (STI) lengths and several interferer power levels. For short STI lengths on the left of the plot, estimation error limits the subspace projection’s null depth. On the right, performance is reduced by subspace smearing. From the results, an optimal range of STI lengths can be identified for a given interferer power level.

**Figure 4 sensors-23-06584-f004:**
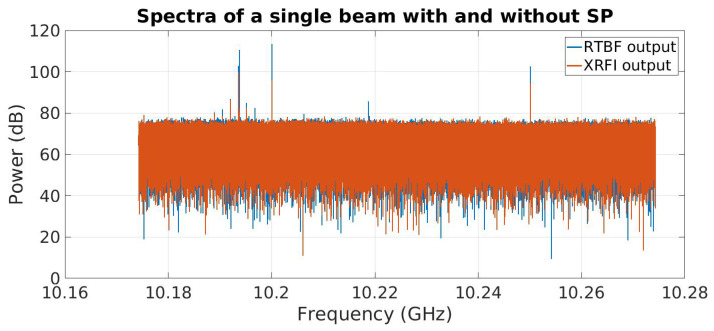
Wideband spectrum plot of beamformer output (RTBF) with no RFI mitigation compared to real-time RFI mitigation (XRFI). Vertical lines indicate the frequencies of tones representing SoI and RFI. The RTBF output shows both tones present while the XRFI mode cancels the RFI using subspace projection.

**Figure 5 sensors-23-06584-f005:**
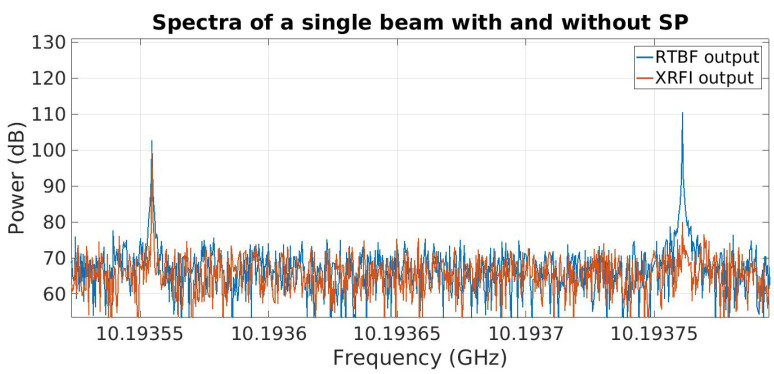
Expanded view of the real-time beamformer (RTBF) output with no RFI mitigation compared to real-time RFI mitigation (XRFI).

**Figure 6 sensors-23-06584-f006:**
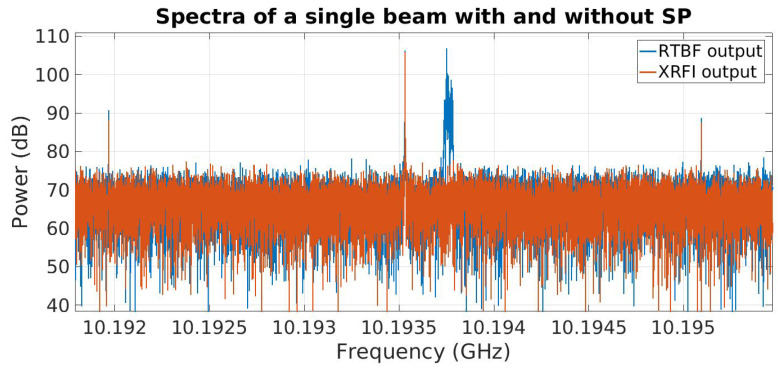
Wideband spectrum of the beamformer output with no RFI mitigation compared to real-time RFI mitigation (XRFI) with a modulated RFI source.

**Figure 7 sensors-23-06584-f007:**
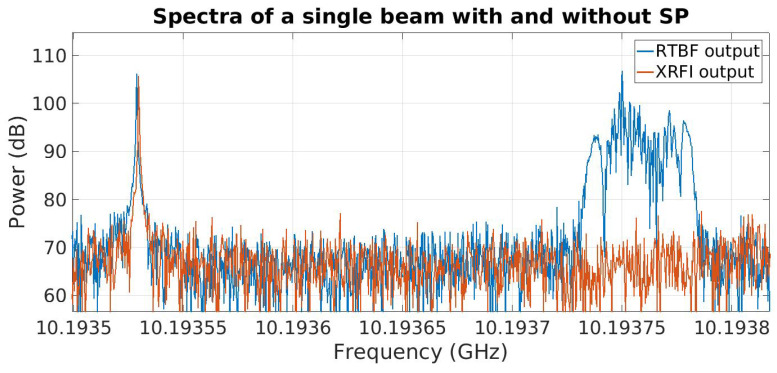
Expanded view of the spectrum of the beamformer output with no RFI mitigation compared to real-time RFI mitigation (XRFI) with modulated RFI source.

## Data Availability

Experimental data unavailable.

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
