# Peer review of "Wideband Array Signal Processing with Real-Time Adaptive Interference Mitigation"

_sensors, 2023, doi:10.3390/s23146584_

Round 1

Reviewer 1 Report (New Reviewer)

Please see the attached comments.

Minor editing of English language is needed.

Author Response

Reviewer 2 Report (New Reviewer)

Article describes design and implementation of high-throughput 16 microstrip patch antenna element phased antenna array with field programmable gate array (FPGA) frequency channelization.  It provides 150 MHz bandwidth and interference cancellation.  Similar systems are used by many astronomic receivers.  Experimental results demonstrate the real-time cancellation of narrow-band interferers within the system operating in wide bandwidth with 10 GHz center frequency. 

Questions:

Why the size of array is 4x4, is it limited by hardware or software, and to what size it can be extended with presented technique?

It would be advantageous if authors mention in abstract and conclusions the impressive numerical result of interference mitigation by 30-35 dB.

Author Response

This manuscript is a resubmission of an earlier submission. The following is a list of the peer review reports and author responses from that submission.

Round 1

Reviewer 1 Report

1. The motivation of the paper is not clear. More rationales for the proposed method can be provided for a better understanding of the paper.  

2. What are the advantages of the developed method compared with existing schemes? 

3. There is a lack of punctuation in the paper. Please double-check this point. 

4. The system model is not formulated very clearly. More details of the system model are important for a better understanding of the paper. 

Please double-check the punctuation of the formulations.  

Reviewer 2 Report

This paper shows a heterogeneous signal processing architecture to adaptive beamforming sensor arrays for wideband communications. There are some problems for this paper:

1、 The authors think that their key contribution is a hardware and software pipeline for subchannelized wideband array signal processing with 150 MHz instantaneous bandwidth and real-time interference cancellation with a heterogeneous, distributed, and scaleable DSP architecture. However, the introduction and explanation of this architecture are very limited in the text. The difficulties and key technologies of the system have not been highlighted, so that readers cannot discover the innovation of the proposed architecture.

2、 The article uses many abbreviations and abbreviations, especially some abbreviations that are not given full names for the first time, and the readability needs to be improved.

3、 The simulation corresponding to Figure 3 does not provide complete simulation conditions, such as signal and interference bandwidth, angle, and signal-to-noise ratio.

4、 The experimental results part requires significant modifications. The two experiments in the existing articles have the following issues:

a)      The signal and interference forms of the first experiment were not provided, and their corresponding parameters such as bandwidth, center frequency, signal-to-noise ratio, and interference to noise ratio were not explained.

b)     The form of the signal in the second experiment was not provided, and the corresponding parameters of the signal, such as bandwidth, center frequency, signal-to-noise ratio, and the interference to noise ratio for the interference, were not explained.

c)      The third problem is the most fatal. In the first experiment, it can be seen from the experimental results that both the signal and interference are in the form of single frequency signal. In the second experiment, although the interference has a certain bandwidth, it is far from meeting the conditions for broadband signals. So both experiments are actually the results of narrowband signal processing, not the true results of broadband signal processing. This does not match the title of the paper.

d)     The results of both experiments indicate the problem of aliased signal. Is it possible to eliminate this phenomenon by making improvements to the system design during signal processing?

 The article uses many abbreviations and abbreviations, especially some abbreviations that are not given full names for the first time, and the readability needs to be improved.

Round 2

Reviewer 2 Report

The system presented in the article is real-time interference suppression for wideband array signal processing. It is generally believed that a signal with a bandwidth to center frequency ratio exceeding 10% can be considered a wideband signal. However, all the experiments presented in the paper were performed with narrowband signals and narrowband interference. From this point of view, the effectiveness of the broadband array signal processing system has not been effectively verified. The effectiveness is not determined, and the contribution of the author's work needs to be discounted.

Author Response

We thank reviewer 2 for looking over the revised manuscript.

Reviewer 2 offers one final remaining objection, that the system has not been demonstrated for a wideband signal.

This critique is based on a misunderstanding. In many applications for array receivers, the noise floor *is* the signal of interest. Figure 4 clearly shows the wideband system output noise floor and demonstrates that the system processes a full 150 MHz wideband signal. The system reported in the paper is certainly processing a wideband signal. 

The narrowband tone representing a CW signal shown in figures is used only for visual convenience, as it can be seen above the noise floor. A wideband test signal would be indistinguishable from the noise floor and there is no good reason to add that to the results. 

Furthermore, wideband signal processing with the heterogenous FPGA/GPA architecture has already been demonstrated and published by many groups (including our own). If we were to add more results demonstrating what has already been published, this would be redundant. The point of our work is to add RFI cancellation to the wideband FPGA/GPA signal processing architecture.

As far as the RFI itself is concerned, the system was never intended to reject wideband RFI. The system is designed to reject narrowband RFI. We have clarified this in the introduction in the second revision. The RFI rejection algorithm operates within the FFT filterbank subchannels that are used in wideband array communications systems and astronomical receivers, which is actually an advantage. We have pointed this out in the conclusion of the second revision of the paper. 

As the experimental goal of the work has been fully accomplished, there is no further work, expansion of the system, or experimentation that could be done to improve upon the reported work. 

Based on these considerations, we encourage the editor to disregard the reviewer's final objection to the paper.